# Experimental Study on Partially Coherent Optical Coherent Detection

**Jingyuan Liang** [1,2], **Yi Mu** [1], **Xizheng Ke** [1,2,*] **and Meimiao Han** [1,2]

1   School of Automation & Information Engineering, Xi'an University of Technology, Xi'an 710048, China; ljy@xaut.edu.cn (J.L.); 2210321157@stu.xaut.edu.cn (Y.M.); 1220311007@stu.xaut.edu.cn (M.H.)
2   Shaanxi Intelligent Collaborative Network Military, Civilian Integration Key Laboratory, Xi'an 710048, China
*   Correspondence: xzke@xaut.edu.cn

**Abstract:** When coherent detection occurs, the polarization mismatch between signal light and local oscillator light can reduce the efficiency of coherent detection. This article combines the principle of optical mixers to derive the relationship between the polarization state and mixing efficiency of signal light and local oscillator light, and builds an experimental platform for the coherent detection of a partially coherent electromagnetic Gaussian Schell beam (EGSM). Polarization devices are used to regulate the polarization state of the signal EGSM light and local oscillator EGSM light, and different polarization states of the EGSM beams are generated. When the output power of the signal light is constant, the mixing efficiency is measured according to the output amplitude of the intermediate frequency signal. This experiment found that when the signal light is in a linearly polarized state and the local oscillator light is in a linearly polarized state, a circularly polarized state, or an elliptically polarized state, the amplitude of the intermediate frequency signal is 369.6 mv, 146.6 mv, or 92.1 mv, respectively. When the signal light is in a circularly polarized state, the amplitude of the intermediate frequency signal is 446.4 mv, 504.0 mv, or 159.2 mv, respectively. When the signal light is in an elliptical polarization state, the amplitude of the intermediate frequency signal is 94.4 mv, 124.0 mv, or 254.8 mv, respectively.

**Keywords:** electromagnetic Gaussian Schell beam; optical mixer; coherent detection; polarization state

## 1. Introduction

In a coherent optical communication system, if the polarization state of the signal light is uncertain [1], but the optical mixer is sensitive to the polarization state of the signal light and the local polarization light, and the polarization state of the signal light does not match with the local polarization light, it will reduce the heterodyne's efficiency and seriously affect the performance of the whole communication system [2]. In coherent optical communication systems, the generation of intermediate frequency signals relies on an optical mixing process, which is the nonlinear interaction between local oscillator light and signal light in the photodetector. This interaction facilitates the frequency transformation of the signal to the intermediate frequency range, facilitating subsequent signal processing in the form of lower-frequency electrical signals. Due to the fact that the mixing rate process depends on the phase relationship between the local oscillator light and the signal light, different polarization states may also introduce phase uncertainty, which can affect the phase of the intermediate frequency signal and thus affect the demodulation quality of the signal. Polarization mismatch may result in some components of the signal not being effectively mixed, which may cause a decrease in signal quality, such as amplitude or phase distortion, further reducing the signal-to-noise ratio (SNR) and bit error rate (BER) performance of the signal.

Hema R [3] used the properties of the cross spectral density matrix of an EGSM source characterized by 10 parameters to derive the necessary and sufficient conditions that must be met to produce this type of physically achievable beam source. Zhang Biling [4] derived

analytical formulas for the root mean square spatial width, root mean square angular width, and $M^2$-factor of EGSM beams in turbulence using the Wigner distribution function and the extended Huygens Fresnel integration second-order moment formula. The results indicate that the relative $M^2$-factor of EGSM beams decreases with a decreasing zenith angle, initial coherence length, and initial polarization degree, as well as an increasing beam width and intrinsic scale. Yang Xianyang [5] investigated the effects of anisotropic atmospheric turbulence on the propagation characteristics of an electromagnetic twisted Gaussian Schell-model array beam. They derived an analytical expression for the cross-spectral density function of the beam propagating through anisotropic turbulent atmosphere and utilized it to explore the evolutionary behavior of the spectral intensity, degree of polarization, and degree of coherence. Sethuraj R [6] has outlined a method based on first-order interference and two-point Stokes parameters for measuring the polarization states of electromagnetic Gaussian Schell-model beams, and has applied second-order field correlations to determine the complex magnitude and phase of their electromagnetic coherence. The effectiveness of this method was demonstrated experimentally using an EGSM beam generated by a laser beam passing through a rotating ground glass diffuser. Yao Min [7] studied the evolution of the degree of polarization of EGSM beams. The interaction between EGSM beams and Gaussian cavities was analyzed. The results indicate that the behavior of the degree of polarization depends on the statistical properties of the source generating the EGSM beams and the parameters of the cavity. Zhao Yuanhang [8] studied the polarization properties of electromagnetic Gaussian Schell-model beams propagating through the anisotropic non-Kolmogorov turbulence of a marine-atmosphere channel based on the cross-spectral density matrix of the beam. The electromagnetic Gaussian Schell-model beam with smaller $\delta yy$, $\delta xx$, and $Ax$ parameters, but a larger $Ay$, will reduce the interference of turbulence.

Tanaka T et al. [9] investigated the behavior of coherent detection systems in the case of partially coherent beams containing both local oscillator light and signal light. Using incoherent mode decomposition theory and the generalized Huygens–Fresnel principle as foundations, Ke Xizheng et al. [4], using the real heterodyne detection system on a 1.3 km outfield communication link, calculated the expression of the weight factor of a number of GSM beams at the receiver. Salem and Rolland [10] investigated how a coherent detection system was affected by the angular error between the signal beam and the partially coherent beam's local oscillator light. A theoretical expression was derived for the heterodyne efficiency of mixing two partially coherent beams with a small angular displacement between their propagation directions. Li [11] addressed the effect of turbulence on a coherent detection system based on partially coherent beam theory. Wang et al. [12] used the Gaussian Schell-model's electromagnetic beam coherent polarization matrix to derive an analytical expression of a coherent detection system. The electromagnetic field's polarization cannot be disregarded, and a beam's polarization typically changes as it propagates. According to research by Ke Xizheng [13], under the same circumstances, the uplink propagation of the EGSM beam has a less concentrated distribution across the degree of polarization of the entire light field than the downlink propagation, and the downlink propagation's transmission distance is longer than the uplink transmission's, corresponding to the maximum polarization on the axis. A detector can obtain the beam transmission data at a greater distance as the EGSM beam moves along the downlink path, as per the non-Kolmogorov turbulence spectrum. Wu Jiali et al. [14] investigated the effects of EGSM beams on the sensitivity of a coherent detection system under various polarization states and derived the expression of the coherent detection system's sensitivity for the oblique propagation of EGSM beams.

In summary, in the theoretical analysis of partially coherent optical heterodyne detection systems, this paper takes coherent optical communication as the background, generates EGSM beams in its experiments, and proposes and practices an innovative experimental method that can accurately prepare and adjust the polarization state of EGSM beams, including their linear polarization, circular polarization, and elliptical polarization EGSM. And this study not only analyzed the polarization state of partially coherent beams, but

also specifically investigated the impact of partially coherent EGSM beams on system performance at different polarization states, with a focus on analyzing the impact of partially coherent light polarization states on mixing efficiency, providing important support for improving EGSM beams' performance in wireless optical communication and other fields.

## 2. Theoretical Analysis

### 2.1. Analysis of the Polarization Characteristics of Partially Coherent Light

Electromagnetic Gaussian Schell-mode beams are a convenient model for studying coherence and polarization theory, and have attracted increasing attention due to their potential applications [15]. The polarization properties of vectored partially coherent beams can be represented by polarization ellipses.

Given that the EGSM beam source is located in the $z > 0$ plane, the beam's cross-spectral density matrix is known to be [16]

$$W_{ij} = (\rho, \rho, z) = \frac{A_i A_j B_{ij}}{\Delta_{ij}^2(z)} \exp\left[-\frac{\rho^2}{2\sigma_i \sigma_j \Delta_{ij}^2(z)}\right] (i = x, y; j = x, y) \tag{1}$$

$\rho$ represents the position vector of any point on the $z > 0$ plane. $A_i$ and $A_j$ represent the amplitudes of the electric field component of the beam in the $i$ direction and $j$ direction, respectively. $B_{ij}$ is a phase correlation factor. $\sigma$ represents the effective width of the spectral density in the $i$ and $j$ directions. The beam spread equals $\Delta_{ij}^2 = 1 + \alpha_{ij}z^2$ $(i = x, y; j = x, y)$, where $\alpha_{ij} = \frac{1}{(k\sigma)^2}\left(\frac{1}{4\sigma^2} + \frac{1}{\delta_{ij}^2}\right)$. The coherent length of the field in various directions on the cross-section is characterized by $\delta_{ij}$. The polarization of the EGSM beam can be characterized using the polarization ellipse, which represents the entire polarization component of the electromagnetic Gaussian Schell beam. It has been proven that the cross-spectral density matrix of an electromagnetic Gaussian Schell beam can be decomposed into its completely polarized part and completely unpolarized part [16].

$$W(\rho, \rho, z) = W^{(u)}(\rho, \rho, z) + W^{(p)}(\rho, \rho, z) \tag{2}$$

where

$$W^{(u)}(\rho, \rho, z) = \begin{pmatrix} A(\rho, \rho, z) & 0 \\ 0 & A(\rho, \rho, z) \end{pmatrix} \tag{3}$$

$$W^{(p)}(\rho, \rho, z) = \begin{pmatrix} B(\rho, \rho, z) & D(\rho, \rho, z) \\ D(\rho, \rho, z) & C(\rho, \rho, z) \end{pmatrix} \tag{4}$$

Here $A(\rho,\rho,z)$, $B(\rho,\rho,z)$, $C(\rho,\rho,z)$, and $D(\rho,\rho,z)$ can be expressed as

$$
\begin{aligned}
A(\rho, \rho, z) =\ &\tfrac{1}{2}[W_{xx}(\rho, \rho, z) + W_{yy}(\rho, \rho, z) \\
&+ \sqrt{[W_{xx}(\rho, \rho, z) - W_{yy}(\rho, \rho, z)]^2 + 4|W_{xy}(\rho, \rho, z)|^2}]
\end{aligned}
\tag{5}
$$

$$
\begin{aligned}
B(\rho, \rho, z) =\ &\tfrac{1}{2}[W_{xx}(\rho, \rho, z) - W_{yy}(\rho, \rho, z) \\
&+ \sqrt{[W_{xx}(\rho, \rho, z) - W_{yy}(\rho, \rho, z)]^2 + 4|W_{xy}(\rho, \rho, z)|^2}]
\end{aligned}
\tag{6}
$$

$$
\begin{aligned}
C(\rho, \rho, z) =\ &\tfrac{1}{2}[W_{yy}(\rho, \rho, z) - W_{xx}(\rho, \rho, z) \\
&+ \sqrt{[W_{xx}(\rho, \rho, z) - W_{yy}(\rho, \rho, z)]^2 + 4|W_{xy}(\rho, \rho, z)|^2}]
\end{aligned}
\tag{7}
$$

$$D(\rho, \rho, z) = W_{xy}(\rho, \rho, z) \tag{8}$$

These variables satisfy a quadratic equation, which can be represented as the polarization ellipse, an elliptic equation.

$$C(\rho,\rho,z)\varepsilon_x^{(r)2}(\rho,\rho,z) - 2\mathrm{Re}D(\rho,\rho,z)\varepsilon_x^{(r)2}(\rho,\rho,z)\varepsilon_y^{(r)2}(\rho,\rho,z) \\ + B(\rho,\rho,z)\varepsilon_y^{(r)2}(\rho,\rho,z) = [\mathrm{Im}D(\rho,\rho,z)]^2 \tag{9}$$

where Re and Im stand for a complex number's real and imaginary parts, respectively.

$$\varepsilon_x^{(r)}(\rho,\rho,z) = \sqrt{B(\rho,\rho,z)}\cos(\omega t + \delta_x) \tag{10}$$

$$\varepsilon_y^{(r)}(\rho,\rho,z) = \sqrt{C(\rho,\rho,z)}\cos(\omega t + \delta_y) \tag{11}$$

$$\delta_y - \delta_x = \arg[D(\rho,\rho,z)] \tag{12}$$

Let $a_1$ and $a_2$ be the major and minor axes of an ellipse, respectively. The expression for the ellipticity $\varepsilon$ is as follows:

$$a_{1,2}^2 = \frac{1}{2}\left[\sqrt{(W_{xx} - W_{yy})^2 + 4\left|W_{xy}(\rho,\rho,z)\right|^2} \\ \pm\sqrt{[W_{xx}(\rho,\rho,z) - W_{yy}(\rho,\rho,z)]^2 + 4[\mathrm{Re}W_{xy}(\rho,\rho,z)]^2}\right]^{1/2} \tag{13}$$

$$\varepsilon(\rho,\rho,z) = \frac{a_2(\rho,\rho,z)}{a_1(\rho,\rho,z)} \tag{14}$$

The "+" and "−" in the equation correspond to the long half axis and the short half axis, respectively. The polarization ellipse can be used to express the polarization characteristic of an electromagnetic Gaussian Schell beam. The ellipticity is characterized by $\varepsilon$; when the polarization state of the EGSM beam is circular, $\varepsilon$ is 1. When the polarization state of the EGSM beam is linear, $\varepsilon$ is 0. When the EGSM beam is elliptically polarized, $0 < \varepsilon < 1$.

### 2.2. Coherent Detection Principle

The sensitivity of direct detection technology is greatly affected by factors such as the noise of photodetectors, so its signal-to-noise ratio is relatively low. In addition, due to its limitations in detection area and response speed, direct detection technology is not very suitable for some high-speed and long-distance transmission fields. In order to achieve quantum noise limitation detection and detect weak light signals, it is necessary to use highly sensitive coherent detection methods. Coherent detection technology can place light detection systems under quantum noise limitations, achieving the efficient detection of weak light signals.

Photodetectors cannot directly obtain the frequency and phase information of light, but interference occurs when two beams of light overlap with each other. By detecting changes in interference intensity, the phase information of light can be inferred, and then the frequency information of that light can be obtained. Therefore, photodetectors utilize interference phenomena to obtain the phase and frequency information of light waves, in order to achieve coherent detection. In Figure 1, the principle diagram of coherent detection is shown. The signal light and local oscillator light are mixed in the optical mixer, and then the photodetector converts the optical signal into an electrical signal. After circuit processing, the required information is extracted [17].

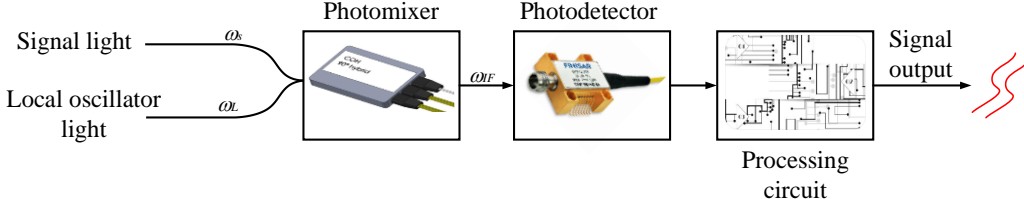

**Figure 1.** Coherent detection principle diagram.

Let the signal light and local oscillator light fields be

$$\begin{cases} E_S = A_S \exp[i(\omega_S t + \varphi_S)] \\ E_L = A_L \exp[i(\omega_L t + \varphi_L)] \end{cases} \tag{15}$$

Here, $A_S$, $\omega_S$, and $\varphi_S$ represent the amplitude, angular frequency, and initial phase of the signal light, respectively; $A_L$, $\omega_L$, and $\varphi_L$ represent the amplitude, angular frequency, and initial phase of the local oscillator light. Assuming that the laser beam sizes of the signal light and the local oscillator light are the same, and the two beams overlap well with each other. The light intensity of the intermediate frequency signal on the detection surface is

$$\begin{aligned} I = \quad & A_S^2 + A_L^2 + A_S A_L \{\exp[i(\omega_S - \omega_L)t - (\varphi_S - \varphi_L))] \\ & + \exp[-i(\omega_S - \omega_L)t - (\varphi_S - \varphi_L))]\} \end{aligned} \tag{16}$$

Simplifying Equation (16) yields

$$I = A_S^2 + A_L^2 + 2A_S A_L \cdot \cos[(\omega_S - \omega_L)t + (\varphi_S - \varphi_L)] \tag{17}$$

where $\Delta\omega = \omega_S - \omega_{LO}$ and $\Delta\varphi = \varphi_S - \varphi_{LO}$, and the intermediate frequency current returned by the detector is

$$i(t) = \Re \cdot \left\{ A_S^2 + A_L^2 + 2A_S A_L \cdot \cos[(\omega_S - \omega_L)t + (\varphi_S - \varphi_L)] \right\} \tag{18}$$

Here, $\Re$ is the responsivity of the detector. From Equation (18), it can be seen that the first two terms in the intermediate frequency current response of the detector are DC terms, while the third term is a signal term. Coherent detectors can detect the intensity, frequency, and phase changes of light waves through this signal term. This detection method is very flexible, and suitable modulation methods can be selected to avoid interference, thus achieving high-precision optical signal detection and measurement. In addition, due to the fact that the energy of local oscillator light is several orders of magnitude higher than that of signal light during the mixing process, the energy of the signal term will increase, thereby improving the sensitivity of the detection.

### 2.3. Polarization States and Mixing Efficiency

To investigate the impact of the polarization state of an EGSM beam on mixing efficiency, the Jones matrix is utilized to simulate the 90° spatial optical mixer principle. It compares the optical mixer's mixing efficiency in various polarization states. Figure 2 displays the optical mixer's schematic diagram [18].

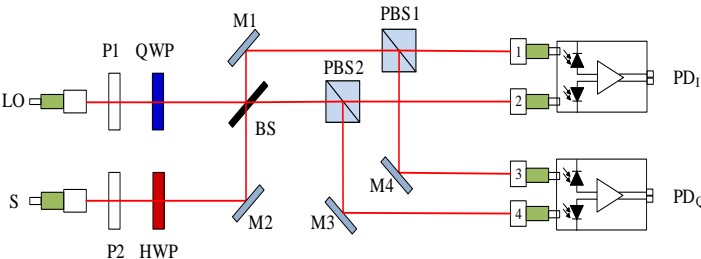

**Figure 2.** Optical path diagram of a 90° optical mixer.

A balanced heterodyne detection schematic diagram is shown in Figure 2. The local oscillator emits a laser beam that first travels through the polarizer P1 with an *x*-axis polarization direction. After that, it travels through a quarter wave plate, where it forms an angle to the *x*-axis of 45 degrees. The signal light laser beam travels through polarizer P2, which is in the same polarization direction as P1, and after passing through a half wave plate, it forms an angle difference of 22.5° with respect to the *x*-axis. Subsequently, the two light beams are divided into four beams via PBS1 and PBS2, respectively, after being

combined through BS. The I and Q electrical signals can then be output using a balanced detector after the four light signals have formed a 90° phase difference.

$E_s$ stands for signal light and $E_{LO}$ for local oscillator light. $k_1$ and $k_2$ represent the polarization component of signal light, $k_3$ and $k_4$ represent the polarization component of local polarized light. $\varphi(t)$ represents the signal light's phase. $\psi$ is the local oscillator light's phase. $\tau_\parallel$ and $\rho_\perp$ represent the phase factor generated after the polarization splitting prism, representing the change in phase. After calculation, the four output results are $I_0$, $I_{90}$, $I_{180}$, and $I_{270}$, respectively.

$$I_0 = \frac{1}{2}\left\{k_2^2|E_s|^2 + k_3^2|E_{LO}|^2 + 2k_2k_3|E_sE_{LO}|\exp\left\{i\left[\psi + \tau_\parallel + \frac{\pi}{4} - \varphi(t) - \rho_\perp\right]\right\}\right\} \quad (19)$$

$$I_{90} = \frac{1}{2}\left\{k_1^2|E_s|^2 + k_4^2|E_{LO}|^2 + 2k_1k_4|E_sE_{LO}|\exp\left\{i\left[\varphi(t) + \tau_\parallel - \psi - \rho_\perp - \frac{\pi}{4}\right]\right\}\right\} \quad (20)$$

$$I_{180} = \frac{1}{2}\left\{k_2^2|E_s|^2 + k_3^2|E_{LO}|^2 - 2k_2k_3|E_sE_{LO}|\exp\left\{i\left[\varphi(t) + \rho_\perp - \psi - \tau_\parallel - \frac{\pi}{4}\right]\right\}\right\} \quad (21)$$

$$I_{270} = \frac{1}{2}\left\{k_1^2|E_s|^2 + k_4^2|E_{LO}|^2 - 2k_1k_4|E_sE_{LO}|\exp\left\{i\left[\psi + \rho_\perp + \frac{\pi}{4} - \varphi(t) - \tau_\parallel\right]\right\}\right\} \quad (22)$$

Based on the four output photocurrents obtained above, the DC signal $I_I$ and AC signal $I_Q$ can be calculated for subsequent information restoration and phase locking, respectively. It is evident that, following passage through a PBS, the two light paths' polarization states (p- and s-polarization) separate, allowing the p-polarized light to be fully transmitted and the s-polarized light to be reflected at a 45-degree angle. Both the local oscillator light and the signal light's vertical and parallel components enter two branches, respectively, creating two mixed beams with perpendicular polarization directions. When $\tau_\parallel$ and $\rho_\perp$ satisfy $\tau_\parallel - \rho_\perp = \pm\frac{\pi}{4}$, a phase delay of 90° can be achieved.

The signal and local oscillator light's incident on the mixer can be expressed as follows, when the signal light is linearly polarized.

$$E_s = E_s(x, y, z)\begin{pmatrix} k_1 \\ k_2 \end{pmatrix}\exp(i\phi(t)) \quad (23)$$

$$E_{LO} = E_{LO}(x, y, z)\begin{pmatrix} k_3 \\ k_4 \end{pmatrix}\exp(i\varphi(t)) \quad (24)$$

The signal light's parallel and vertical polarization components are denoted by $k_1$ and $k_2$, while the local oscillator light's parallel and vertical polarization components are represented by $k_3$ and $k_4$, which, respectively, fulfill the equations $k_1{}^2 + k_2{}^2 = 1$ and $k_3{}^2 + k_4{}^2 = 1$. The signal light's phase modulation is represented by $\phi(t)$ and the local oscillator light's phase is represented by $\varphi(t)$.

Following the transmission of the local oscillator light and the linearly polarized signal light to the optical mixer, the output of the orthogonal branch photocurrent, for loop control, and the DC branch photocurrent, for restoring modulation information, can be expressed as follows:

$$I_I = 2k_2k_3|E_sE_{LO}|\cos\phi(t) \quad (25)$$

$$I_Q = 2k_1k_4|E_sE_{LO}|\cos\left(\phi(t) + 2(\tau_\parallel - \rho_\perp) + \frac{\pi}{2}\right) \quad (26)$$

It is evident from the photocurrents of the *I* and *Q* channels that the polarization components of the signal light and local oscillator light, denoted as $k_1$, $k_2$, $k_3$, and $k_4$, will influence the output photocurrent's intensity, thereby impacting the output signal's amplitude and the power output of the mixer, altering the mixing efficiency and the signal light's energy distribution in the process.

Equations (25) and (26), which simplify to yield two input optical signals, can be simplified by substituting the polarization components $k_1$, $k_2$, $k_3$, and $k_4$ with $\frac{\sqrt{2}}{2}$, in an ideal scenario (where the angle $\theta$ between the fast axis of the wave plate is 45°).

$$I_I = \sqrt{2}\sin\theta |E_s E_{LO}| \cos\phi(t) \tag{27}$$

$$I_Q = \sqrt{2}\cos\theta |E_s E_{LO}| \cos\left(\phi(t) + \frac{\pi}{2}\right) \tag{28}$$

where $\theta$ is the angle formed between the horizontal direction and the light vector. The angle $q$ only modifies the light's energy distribution, according to the analysis in the preceding text. Thus, efficient mixing is possible when there is an ideal phase difference, i.e., when the $I$ and $Q$ phases stay synchronized.

Output power is taken into consideration when examining the impact of the angle $q$ between the horizontal direction and the optical vector on mixing efficiency. It is possible to express the mixing efficiency as the ratio between the output signal's orthogonal and in-phase component amplitudes. By assuming that the input signal's in-phase and orthogonal components are represented as $I_{Iin}$ and $I_{Qin}$, respectively, and the output signal's in-phase and orthogonal components are represented as $I_{Iout}$ and $I_{Qout}$, respectively, the mixing efficiency can be calculated as follows:

$$\eta = \frac{I_{Iout}^2 + I_{Qout}^2}{I_{Iin}^2 + I_{Qin}^2} \tag{29}$$

Formula (29) can be simplified by substituting Formulas (27) and (28) into it.

$$\eta(\theta) = \frac{\left(\sqrt{2}\sin\theta + \sqrt{2}\cos\theta\right)^2}{4} = \frac{|1 + \sin 2\theta|}{2} \tag{30}$$

The spatial optical mixer's mixing efficiency can be calculated using the mathematical model formula [19].

$$\eta_c = \frac{\left[\int_0^{r_0} E_s(r)E_{LO}(r)\exp(i\Delta\varphi)rdr\right]^2}{\int_0^{r_0}[E_{LO}(r)]^2 rdr \int_0^{r_0}[E_s(r)]^2 rdr} \xrightarrow{\substack{\text{Energy change.} \\ \text{and phase matching.}}} \frac{\left[\int_0^{r_0}\eta(\theta)E_s(r)E_{LO}rdr\right]^2}{\int_0^{r_0}[E_{LO}(r)]^2 rdr \int_0^{r_0}[E_s(r)]^2 rdr} \tag{31}$$

The radius of the detector is denoted by $r_0$. The Formula (31) indicates that the mixing efficiency can be expressed by Formula (30), provided that the phase difference deviation is disregarded and both light fields are perfectly matched.

Figure 3 displays the simulation calculation of Formula (30). Figure 3 shows that the maximum optical power of the output signal occurs when the angles of the signal light vector and the local vibrating light vector are 45 degrees with respect to the horizontal direction. As a result, the mixing efficiency increases by the greatest amount. Accordingly, the maximum mixing efficiency occurs when the electromagnetic Gaussian Schell beam is linearly polarized and the angle between the horizontal direction and the light vector is 45°.

With the use of a quarter wave plate, the polarization state of the beam is altered in order to investigate the impact of both circular and elliptical polarization on the operation of the optical mixer.

One feature of the quarter wave plate is its ability to modify the polarization state of light without influencing its intensity or signal power. Consequently, by altering the angle $\theta$ between the fast axis of the plate, a quarter wave plate can be added at the incident end to replicate various polarization states of the signal light and to assess and confirm the intensity of the optical mixer's output signal.

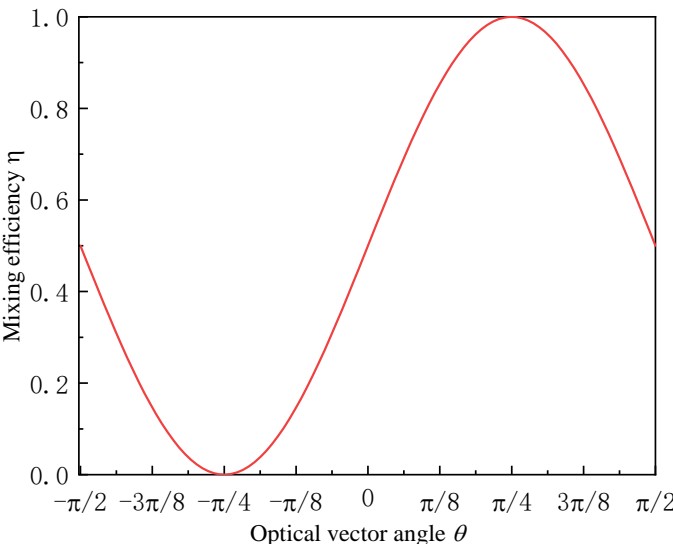

**Figure 3.** Mixing efficiency variation with the light vector angle of linear polarization signal.

A quarter wave plate was chosen to be added to the optical path of the signal light incident on the polarization splitting prism in the original design of the ninety degree mixer. For subsequent calculations, the added quarter wave plate will be represented as

$$\Lambda_{1/4} = \frac{\sqrt{2}}{2} \begin{pmatrix} 1 - i\cos(2\theta) & -i\sin(2\theta) \\ -i\sin(2\theta) & 1 + i\cos(2\theta) \end{pmatrix} \tag{32}$$

The signal and local oscillator light that is incident on the mixer can be written as

$$E_s' = \Lambda_{1/4}E_s = \frac{\sqrt{2}}{2} \begin{pmatrix} 1 - i\cos(2\theta) & -i\sin(2\theta) \\ -i\sin(2\theta) & 1 + i\cos(2\theta) \end{pmatrix} \begin{pmatrix} k_1 E_s e^{i\phi(t)} \\ k_2 E_s e^{i\phi(t)} \end{pmatrix} \tag{33}$$

$$E_{LO}' = \begin{pmatrix} k_3 E_{LO} \exp(i(\frac{\pi}{4} + \psi)) \\ k_4 E_{LO} \exp(i(\frac{\pi}{4} + \psi)) \end{pmatrix} \tag{34}$$

When the beam passes through a polarization splitting prism, its reflection and transmittance are considered ideal. The horizontal and vertical components of the signal light and the local oscillator light are separated and sent into the orthogonal and DC channels. $E_1$ and $E_2$ indicate that the beams passing through the two channels are

$$E_1 = \begin{pmatrix} k_3 E_{Lo} e^{i\tau} e^{i\psi} e^{i\frac{\pi}{4}} \\ -\frac{\sqrt{2}}{2} i k_1 E_s e^{i\rho} e^{i\varphi(t)} \sin(2\theta) + \frac{\sqrt{2}}{2} k_2 E_s e^{i\rho} e^{i\varphi(t)} (1 + i\cos(2\theta)) \end{pmatrix} \tag{35}$$

$$E_2 = \begin{pmatrix} \frac{\sqrt{2}}{2} k_1 E_s e^{i\tau} e^{i\varphi(t)} (1 - i\cos(2\theta)) - \frac{\sqrt{2}}{2} k_2 E_s e^{i\tau} e^{i\varphi(t)} \sin(2\theta) \\ k_4 E_{Lo} e^{i\rho} e^{i\psi} e^{-i\frac{\pi}{4}} \end{pmatrix} \tag{36}$$

Then, passing through a half wave plate with a fast axis direction and a horizontal angle of 22.5°, they become

$$E_1' = \Lambda_{1/2}E_1 = \begin{pmatrix} -\frac{1}{2} i k_1 E_s e^{i\rho} e^{i\varphi(t)} \sin(2\theta) + \frac{1}{2} k_2 E_s e^{i\rho} e^{i\varphi(t)} (1 + i\cos(2\theta)) + \frac{\sqrt{2}}{2} k_3 E_{Lo} e^{i\tau} e^{i\psi} e^{i\frac{\pi}{4}} \\ \frac{1}{2} i k_1 E_s e^{i\rho} e^{i\varphi(t)} \sin(2\theta) - \frac{1}{2} k_2 E_s e^{i\rho} e^{i\varphi(t)} (1 + i\cos(2\theta)) + \frac{\sqrt{2}}{2} k_3 E_{Lo} e^{i\tau} e^{i\psi} e^{i\frac{\pi}{4}} \end{pmatrix} \tag{37}$$

$$E_2' = \Lambda_{1/2}E_1 = \begin{pmatrix} \frac{1}{2} k_1 E_s e^{i\tau} e^{i\varphi(t)} (1 - i\cos(2\theta)) - \frac{1}{2} i k_2 E_s e^{i\tau} e^{i\varphi(t)} \sin(2\theta) + \frac{\sqrt{2}}{2} k_4 E_{Lo} e^{i\rho} e^{i\psi} e^{-i\frac{\pi}{4}} \\ \frac{1}{2} k_1 E_s e^{i\tau} e^{i\varphi(t)} (1 - i\cos(2\theta)) - \frac{1}{2} i k_2 E_s e^{i\tau} e^{i\varphi(t)} \sin(2\theta) - \frac{\sqrt{2}}{2} k_4 E_{Lo} e^{i\rho} e^{i\psi} e^{-i\frac{\pi}{4}} \end{pmatrix} \tag{38}$$

where $\Lambda_{1/2}$ is the Jones matrix of the half wave plate.

$$\Lambda_{1/2} = \frac{\sqrt{2}}{2}\begin{pmatrix} 1 & 1 \\ 1 & -1 \end{pmatrix} \tag{39}$$

Using a polarizing beam splitter prism, output light with a 90° phase difference is generated again.

$$E_0 = -\frac{1}{2}ik_1E_s\sin(2\theta)e^{i(\rho+\varphi(t)+\tau)} + \frac{1}{2}k_2E_s(1+i\cos(2\theta))e^{i(\rho+\varphi(t)+\tau)} + \frac{\sqrt{2}}{2}k_3E_{Lo}e^{i(\psi+2\tau+\frac{\pi}{4})} \tag{40}$$

$$E_{90} = \frac{1}{2}k_1E_s(1-i\cos(2\theta))e^{i(\varphi(t)+2\tau)} - \frac{1}{2}ik_2E_s\sin(2\theta)e^{i(\varphi(t)+2\tau)} + \frac{\sqrt{2}}{2}k_4E_{Lo}e^{i(\psi+2\tau-\frac{\pi}{4})} \tag{41}$$

$$E_{180} = -\frac{1}{2}ik_1E_s\sin(2\theta)e^{i(2\rho+\varphi(t))} - \frac{1}{2}k_2E_s(1+i\cos(2\theta))e^{i(\rho+\varphi(t)+\tau)} + \frac{\sqrt{2}}{2}k_3E_{Lo}e^{i(\psi+\rho+\tau+\frac{\pi}{4})} \tag{42}$$

$$E_{270} = \frac{1}{2}k_1E_s(1-i\cos(2\theta))e^{i(\varphi(t)+\tau+\rho)} - \frac{1}{2}ik_2E_s\sin(2\theta)e^{i(\rho+\varphi(t)+\tau)} - \frac{\sqrt{2}}{2}k_4E_{Lo}e^{i(\psi+2\rho-\frac{\pi}{4})} \tag{43}$$

In path $I$ and path $Q$, the phase information is replaced by parameters $x$ and $y$, respectively; where $x = \varphi(t) - \psi + \rho - \tau - \frac{\pi}{4}$ and $y = \varphi(t) - \psi - \rho + \tau + \frac{\pi}{4}$, and the photocurrent of the four signal light and local oscillator light output beams following coherent mixing can be calculated using the Euler formula, as follows:

$$\begin{aligned} I_0 &= \frac{1}{4}\Big\{[(k_1\sin(2\theta)-k_2\cos(2\theta))^2+k_2^2]|E_s|^2 + 2k_3|E_{Lo}|^2 \\ &\quad +2\sqrt{2}[k_1k_3\sin(2\theta)\sin x + k_2k_3\cos x + k_2k_3\cos(2\theta)\sin(-x)]|E_sE_{Lo}|\Big\} \end{aligned} \tag{44}$$

$$\begin{aligned} I_{90} &= \frac{1}{4}\Big\{[(k_1\cos(2\theta)+k_2\sin(2\theta))^2+k_1^2]|E_s|^2 + 2k_4|E_{Lo}|^2 \\ &\quad +2\sqrt{2}[k_2k_4\sin(2\theta)\sin y + k_1k_4\cos y + k_1k_4\cos(2\theta)\sin y]|E_sE_{Lo}|\Big\} \end{aligned} \tag{45}$$

$$\begin{aligned} I_{180} &= \frac{1}{4}\Big\{[(k_1\sin(2\theta)-k_2\cos(2\theta))^2+k_2^2]|E_s|^2 + 2k_3|E_{Lo}|^2 \\ &\quad +2\sqrt{2}[k_2k_3\cos(2\theta)\sin x - k_2k_3\cos x + k_1k_3\sin(2\theta)\sin(-x)]|E_sE_{Lo}|\Big\} \end{aligned} \tag{46}$$

$$\begin{aligned} I_{270} &= \frac{1}{4}\Big\{[(k_1\cos(2\theta)+k_2\sin(2\theta))^2+k_1^2]|E_s|^2 + 2k_4|E_{Lo}|^2 \\ &\quad +2\sqrt{2}[k_1k_4\cos(2\theta)\sin(-y) + k_2k_4\sin(2\theta)\sin(-y) - k_1k_4\cos y]|E_sE_{Lo}|\Big\} \end{aligned} \tag{47}$$

Subtracting the photocurrent of the 0° and 180° branches obtains the DC signal $I_I'$.

The photocurrents of the 90° and 270° branches can be subtracted to obtain an orthogonal signal $I_Q'$. Then, $I_I'$ and $I_Q'$ can be represented as

$$I_I' = I_0 - I_{180} = \sqrt{2}[k_2k_3\cos x + k_3\sin x(k_1\sin 2\theta - k_2\cos 2\theta)]|E_sE_{LO}| \tag{48}$$

$$I_Q' = I_{90} - I_{270} = \sqrt{2}[k_1k_4\cos y + k_4\sin y(k_1\sin 2\theta + k_2\cos 2\theta)]|E_sE_{LO}| \tag{49}$$

When ignoring external limiting factors such as channel conditions, encoding methods, and the randomness of atmospheric transmission, and only considering the impact of the polarization state changes of the signal light on mixing efficiency, assuming that the linearly polarized EGSM light enters the mixer in an ideal state, with a polarization vector angle of 45°, the linearly polarized EGSM light is uniformly distributed in both horizontal and vertical directions. Therefore, substituting $k_1 = k_2 = k_3 = k_4 = \frac{\sqrt{2}}{2}$ yields

$$\begin{aligned} I_I' &= I_0 - I_{180} = \sqrt{2}[\frac{1}{2}\cos(\varphi(t)-\psi) \\ &\quad +\frac{\sqrt{2}}{2}\sin(\varphi(t)-\psi)\left(\frac{\sqrt{2}}{2}\sin(2\theta)-\frac{\sqrt{2}}{2}\cos(2\theta)\right)]|E_sE_{LO}| \end{aligned} \tag{50}$$

$$
\begin{aligned}
I'_Q \quad &= I_{90} - I_{270} = \sqrt{2}[\tfrac{1}{2}\cos(\varphi(t)-\psi) \\
&+ \tfrac{\sqrt{2}}{2}\sin(\varphi(t)-\psi)\left(\tfrac{\sqrt{2}}{2}\cos(2\theta)+\tfrac{\sqrt{2}}{2}\sin(2\theta)\right)]|E_s E_{LO}|
\end{aligned}
\tag{51}
$$

Taking $\theta$ as a variable between 0 and $\frac{\pi}{2}$ can simulate the situation of signal light with different polarization states imposed on it. The ratio of the amplitude of the in-phase component and the orthogonal component of the output signal is how the mixing efficiency is expressed in the case of an ideal phase difference, as per Formula (25). The mixing efficiency formula is expressed in terms of the output power ratio [18] when looking at output power from the standpoint of its effect on mixer performance.

$$
\eta(\varphi(t)-\psi,\theta) = \frac{[\cos(\varphi(t)-\psi)+\sin(\varphi(t)-\psi)\sin(2\theta)]^2}{2}
\tag{52}
$$

Consequently, Formula (48) can be used to analyze and simulate the effect of signal polarization on mixing performance, as seen from the standpoint of output power. Figure 4 illustrates that the signal light entering the mixer is in a circularly polarized state when there is a 45-degree angle between the fast axis of the quarter wave plate and gravity. At this point, the output signal power reaches its maximum, meaning that the relative mixing efficiency reaches peak 1. The signal light becomes elliptically polarized when the quarter wave plate's fast axis is positioned at a different angle, and, as this angle increases, the mixing efficiency decreases as a result of frequency variations. Therefore, circularly polarized light can be thought of as an optical signal carrier in space laser communication systems in order to achieve efficient mixing.

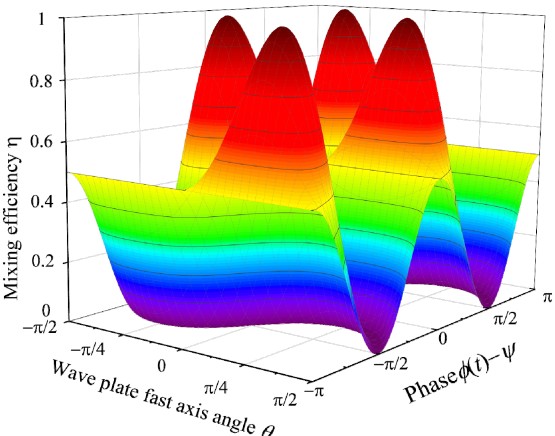

**Figure 4.** Relationship between mixing efficiency, circular polarization, and elliptical polarization.

## 3. Experimental Study

### 3.1. Experimental Setup

Figure 5 illustrates the layout of an experimental schematic system that was constructed indoors to confirm the impact of partially coherent EGSM beams on system performance with various polarization states. The components that make up the system are as follows: a laser, polarizer, transmitting/receiving antenna, 90-degree mixer, oscilloscope, rotating ground glass, convex lens, beam splitter, etc.

Figure 5's optical path indicates how the local oscillator light and signal light are set up, respectively. The laser used in this experiment has an output wavelength of 1550 nm. During the optical path construction process for the experiment, a 650 nm laser with a visible light wavelength is first used to cross the axis, because light at the former wavelength is not visible to the human eye. Next, a precise two-dimensional adjustment frame made of azimuth and elevation adjustments is used to move the laser forward and backward along the beam propagation axis. After collimation, an observation screen is placed behind the laser and moved back and forth to observe the changes in the spot. It is deemed successful

collimation and the rough adjustment is finished if there is virtually no change in the spot diameter over a wide range.

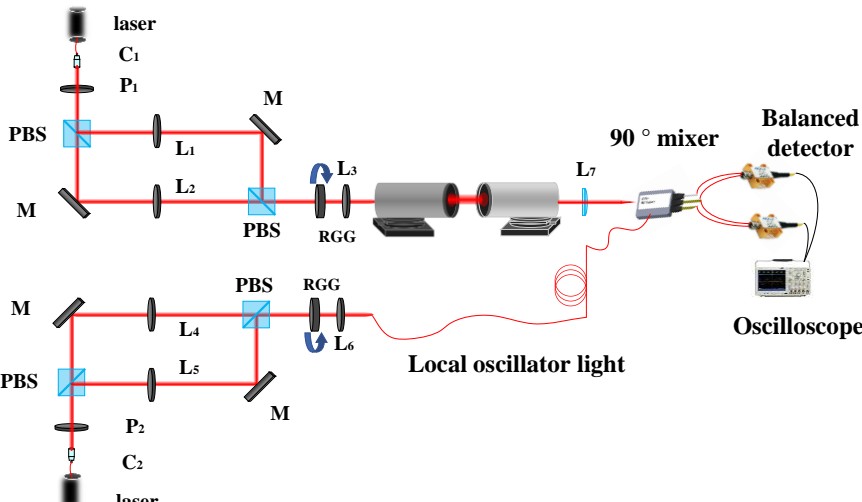

**Figure 5.** Experimental schematic diagram. C1, C2: collimators; P1, P2: polarizers; PBS, polarizing beam splitter; M, mirror; L1–L7, lens; RGG, rotating ground glass.

As shown in Figure 6, after passing through the linear polarizer $P_1$, the laser passes through the Mach Zehnder interferometer. At the output of the Mach Zehnder interferometer, two mutually perpendicular polarized beams are combined and subsequently directed towards a rotating ground glass plate. This interaction results in the generation of a random electromagnetic beam. Lens 3 effectively alters the intensity distribution of the uniform light spot emitted from the ground glass, resulting in the creation of an EGSM beam with an approximate Gaussian distribution. The linear polarizer $P_1$'s transmission direction and the *x*-axis form an angle, and modifying the angle modifies the amplitude ratio of the field Ax/Ay.

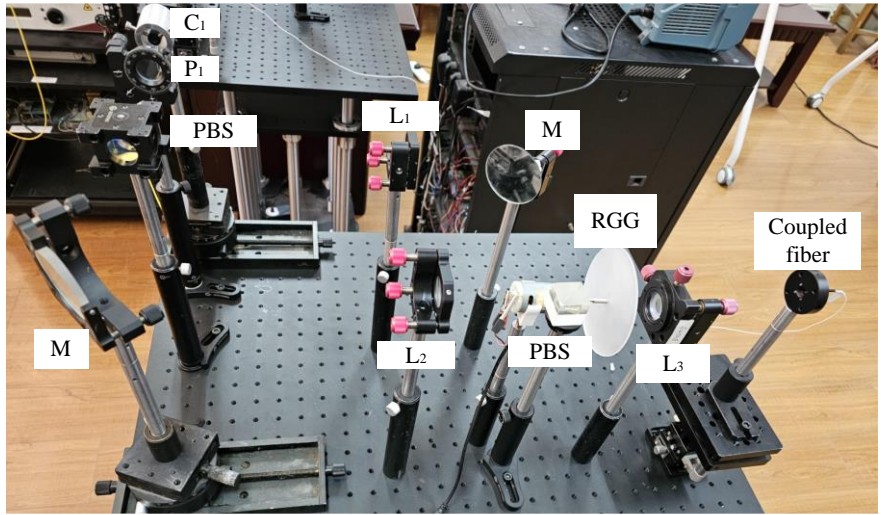

**Figure 6.** Experimental setup for generating EGSM signal light.

The size of the light spot projected onto the rotating ground glass sheet is adjusted using $L_1$ and $L_2$. Both the size of the light spot on the ground glass sheet and the roughness of the ground glass determine the size of $\delta_{ij}$. The process of creating ground glass involves gradually pulverizing regular glass using diamond sand, and the size of the diamond sand particles largely dictates the surface roughness of the glass. The coherence of the beam is not

significantly affected by the ground glass's rotational speed, according to research [20,21]. To construct the optical path for the local oscillator, the same process was used, As shown in Figure 7. The specific parameters of the experimental equipment are shown in Table 1.

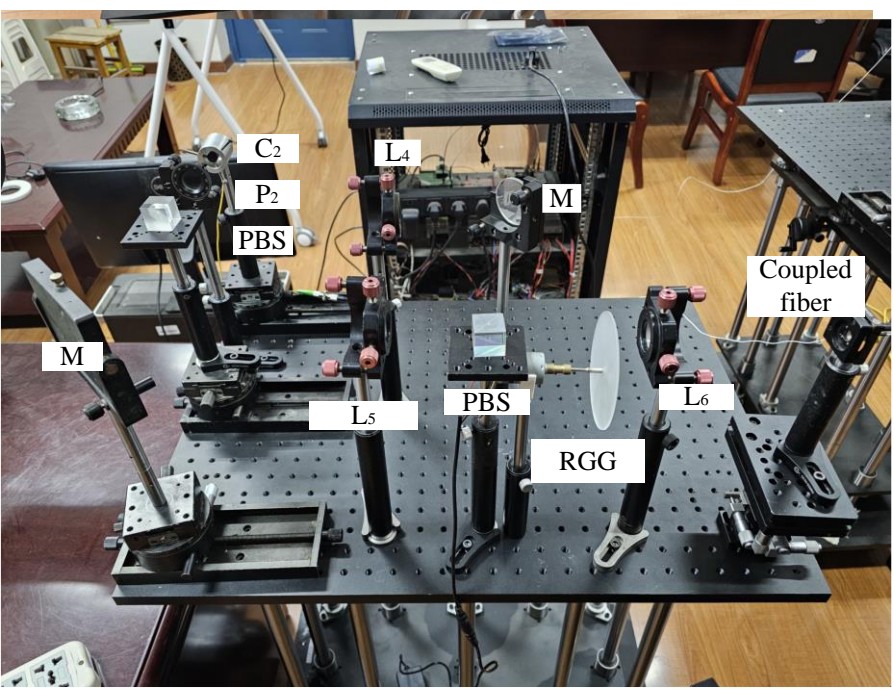

**Figure 7.** Experimental setup for generating EGSM local oscillator light.

**Table 1.** Experimental Equipment.

| Experimental Equipment | Performance Parameter |
| --- | --- |
| Signal laser | Wavelength 1550 nm, line width 0.1 kHz, output power 0–20 mW |
| Local oscillator laser | Wavelength 1550 nm, line width 10 kHz, output power 20 mW |
| Transmitting antenna | Effective aperture 105 mm, obstruction ratio 0.2 |
| Receiving antenna | Effective aperture 105 mm, obstruction ratio 0.2 |
| 90° mixer | Wavelength range 1520–1570 nm, operating temperature 0–70 °C |
| Balanced detector | Wavelength range 1260–1650 nm, response bandwidth 200 MHz |

### 3.2. Partial Coherent Polarization Measurement and the Coherent Detection Experiment

This experiment used optical components like quarter wave plates and polarizers to create signal light and local oscillator light with various polarization states. To achieve an ideal mixing efficiency, good polarization matching must be maintained between the signal light and the local oscillator light.

Local oscillator light with linearly polarized, circularly polarized, and elliptically polarized states was generated using a linearly polarized plate and a quarter wave plate, when the polarization state of the signal light was linear. Using a BOSA 400C spectrometer(Zaragoza, Spain), the various polarization states of signal light and local oscillator light were measured, as indicated in Figure 8.

The data points for the measured signal light and the local oscillator light are both close to the equator of the Poincare sphere, as shown in Figure 8a,b, demonstrating the distribution of linear polarization. As Figure 8c shows that when the polarization characteristics of the beam are controlled with a quarter wave plate, the data point of the local oscillator is close to the south pole of the Banga sphere. The distribution of circular polarization is consistent with this observation. This indicates that the quarter wave plate was successful in changing the polarization state of the beam and creating a circularly

polarized distribution characteristic. The polarization state of the local oscillator is elliptical polarization in Figure 8d.

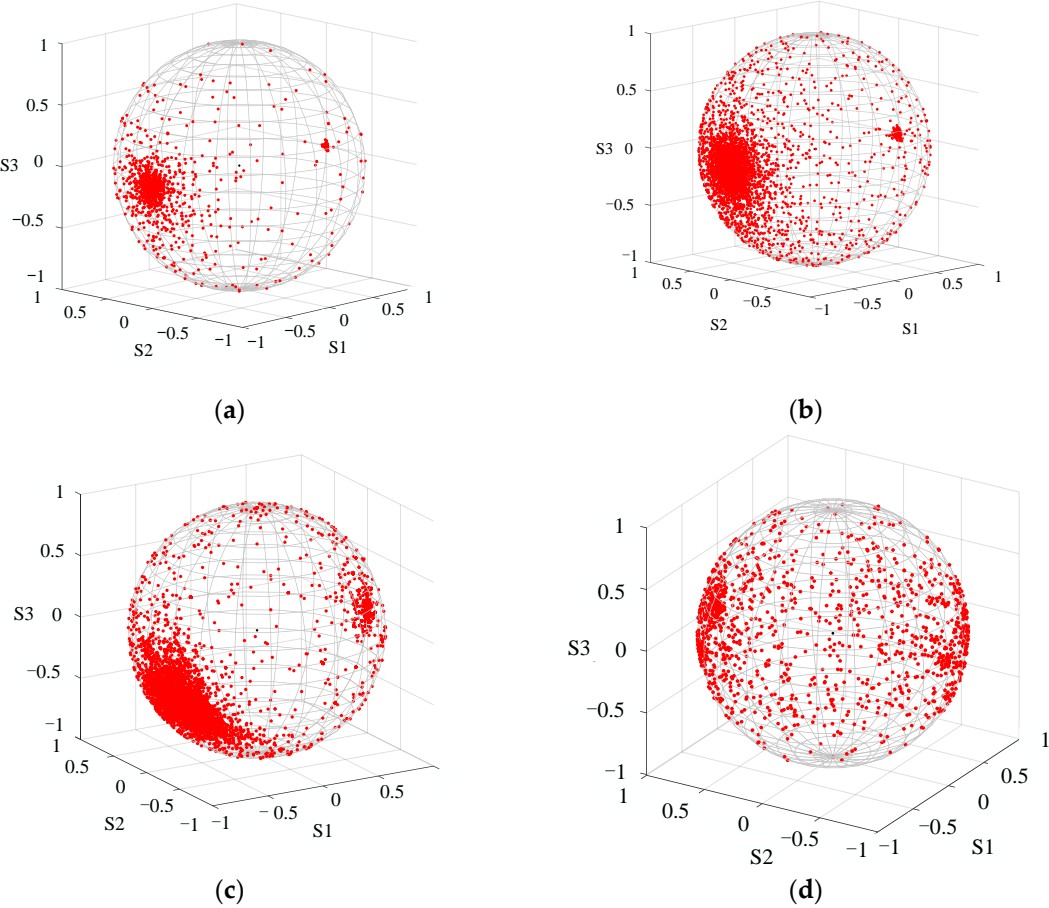

**Figure 8.** Linearly polarized signal light and three polarization states of local oscillator light: linear polarization, circular polarization, and elliptical polarization. (**a**) Linearly polarized signal light; (**b**) linearly polarized local oscillator light; (**c**) circularly polarized local oscillator light; and (**d**) elliptically polarized local oscillator light.

When the signal light's EGSM beam is linearly polarized, the output power of the signal light laser was adjusted to about 1.3 μW, and the output power of the local oscillator laser was fixed at 1.5 mW. The intermediate frequency signal waveform obtained using the Tektronix DPO5204B oscilloscope is displayed in Figure 9.

From the interface of the oscilloscope, it can be seen that when the signal light is in a linearly polarized state and the local oscillator light is linearly polarized, the peak-to-peak value of the intermediate frequency signal is 369.6 mV; when the signal light is linearly polarized and the local oscillator light is circularly polarized, the peak-to-peak value of the intermediate frequency signal is 146.6 mV; and when the signal light is in a linearly polarized state and the local oscillator light is elliptically polarized, the peak-to-peak value of the intermediate frequency signal is 92.1 mV.

Applying the same technique can produce local oscillator light with varying polarization states when the EGSM beam signal is circularly polarized. From Figure 10a it can be seen that the data point of the signal light is close to the south pole of the Banga sphere, which agrees with the distribution of circularly polarized beams on the Banga sphere. Similarly, from Figure 10b–d, it can be seen that circularly polarized local oscillator light, linearly polarized local oscillator light, and elliptically polarized local oscillator light are, respectively, generated.

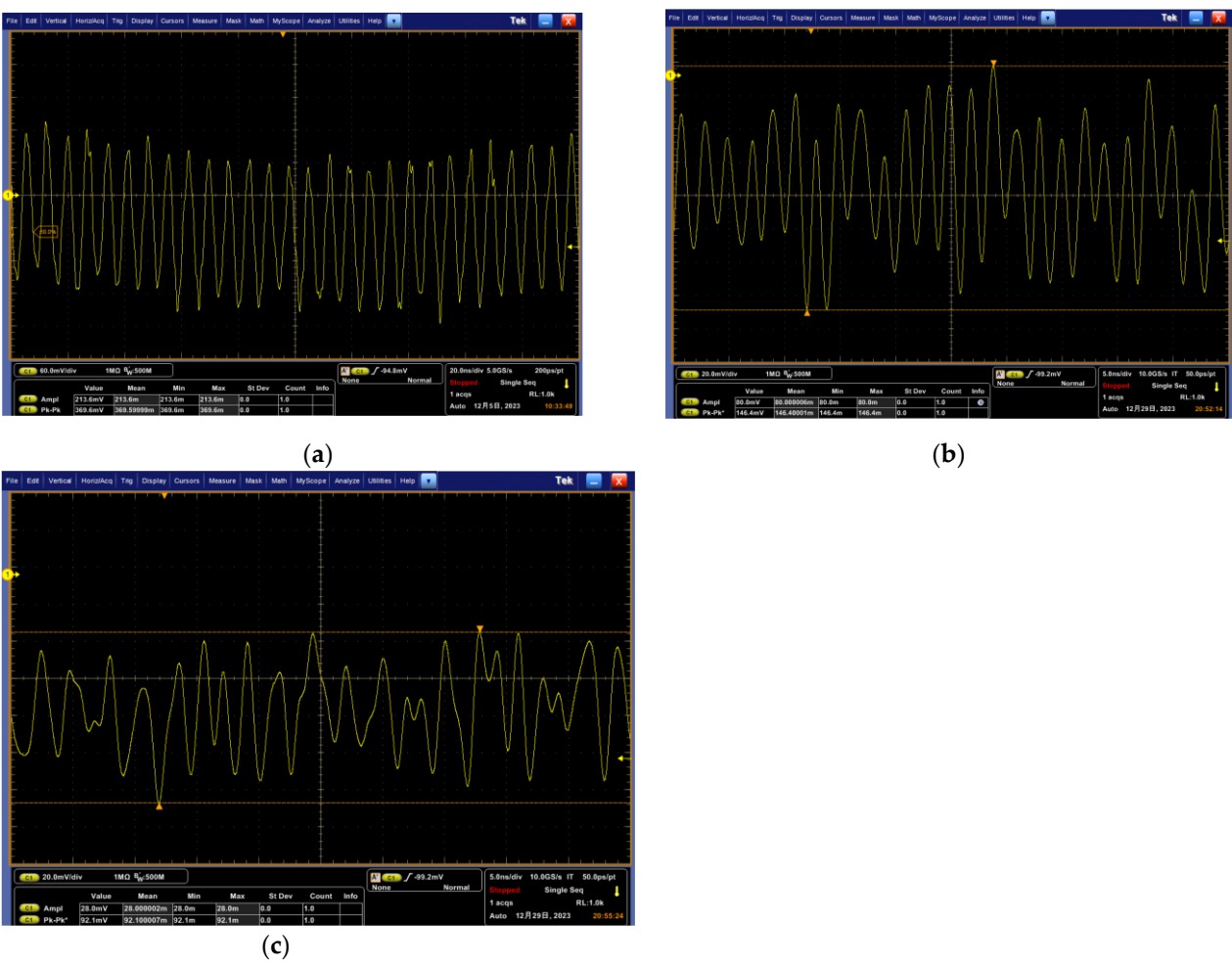

(a)

(b)

(c)

**Figure 9.** Intermediate frequency signal waveforms: (**a**) intermediate frequency signal diagram when the signal light is linearly polarized and the local oscillator light is linearly polarized; (**b**) intermediate frequency signal diagram when the signal light is linearly polarized and the local oscillator light is circularly polarized; and (**c**) intermediate frequency signal diagram when the signal light is linearly polarized and the local oscillator light is elliptically polarized.

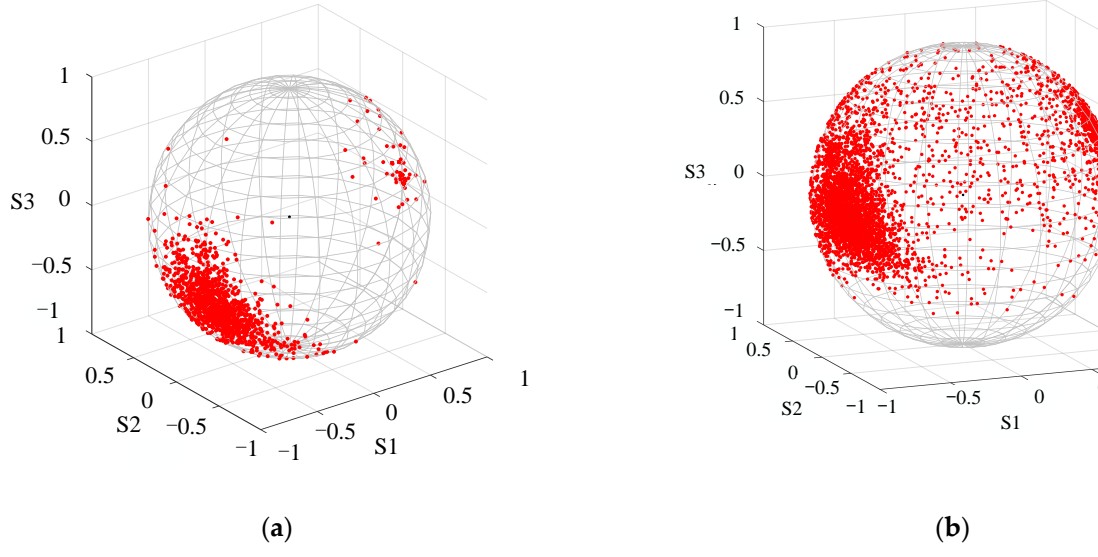

(a)

(b)

**Figure 10.** *Cont.*

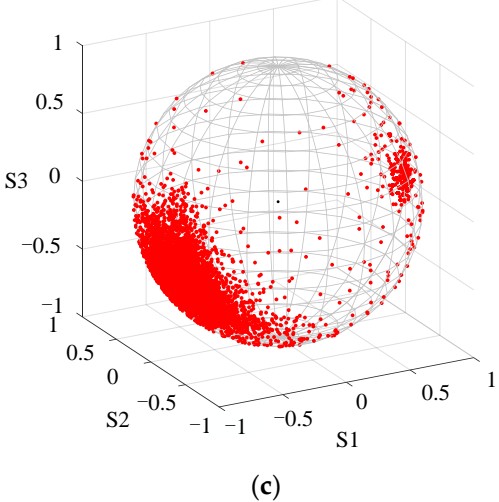

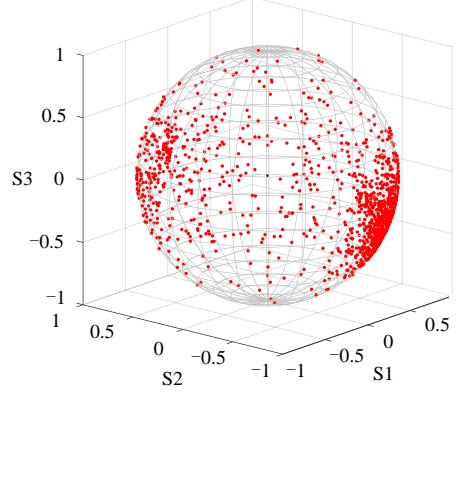

(**c**)                                                                 (**d**)

**Figure 10.** Circularly polarized signal light and three polarization states of local oscillator light: linear polarization, circular polarization, and elliptical polarization. (**a**) Circularly polarized signal light; (**b**) linearly polarized local oscillator light; (**c**) circularly polarized local oscillator light; and (**d**) elliptically polarized local oscillator light.

When the output power of the local oscillator laser is set to 1.5 mW, the power of the signal laser is modified, and the EGSM beam is circularly polarized, the power of the signal light is 1.5 μW.

As shown in Figure 11, the waveform of the intermediate frequency signal collected by the oscilloscope shows that when the signal light is in a circularly polarized state and the local oscillator light is linearly polarized, the peak-to-peak value of the intermediate frequency signal is 446.4 mV; when the signal light is in a circularly polarized state and the local oscillator light is circularly polarized, the peak-to-peak value of the intermediate frequency signal is 504.0 mV; and when the signal light is in a circularly polarized state and the local oscillator light is elliptically polarized, the peak-to-peak value of the intermediate frequency signal is 159.2 mV.

Similarly, following the same operation, elliptically polarized signal light was generated, as shown in Figure 12a. Concurrently, the local oscillator light was engineered to be linearly polarized, circularly polarized, and elliptically polarized, respectively, as shown in Figure 12b–d. The output power of the local oscillator laser was set to 1.5 mW and the power of the signal laser was modified when the EGSM beam was elliptically polarized. At this time, the power of the signal light was 1μW.

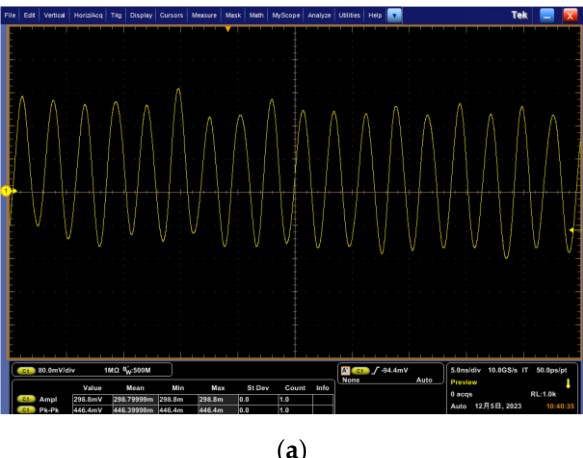

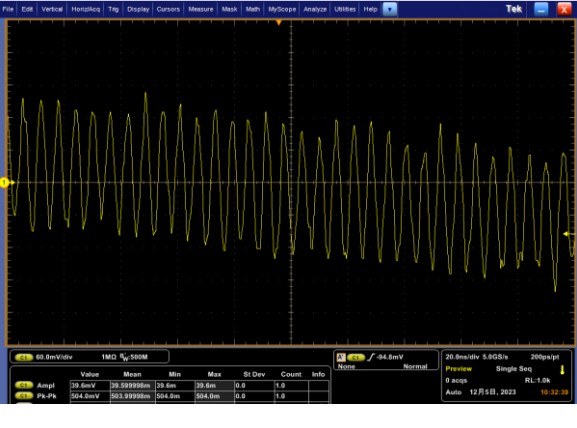

(**a**)                                                                 (**b**)

**Figure 11.** *Cont.*

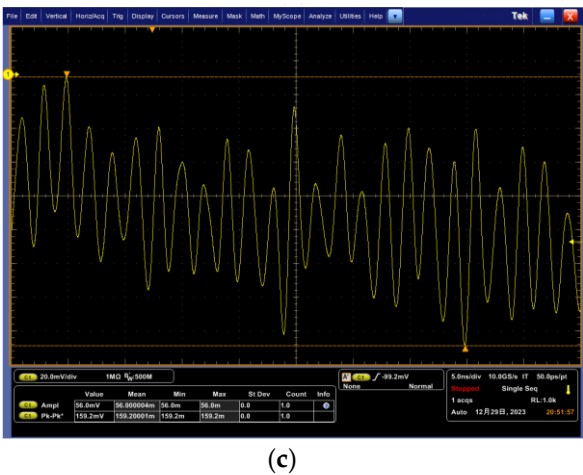

(**c**)

**Figure 11.** Intermediate frequency signal waveforms: (**a**) intermediate frequency signal diagram when the signal light is circularly polarized and the local oscillator light is linearly polarized; (**b**) intermediate frequency signal diagram when the signal light is circularly polarized and the local oscillator light is circularly polarized; and (**c**) intermediate frequency signal diagram when the signal light is circularly polarized and the local oscillator light is elliptically polarized.

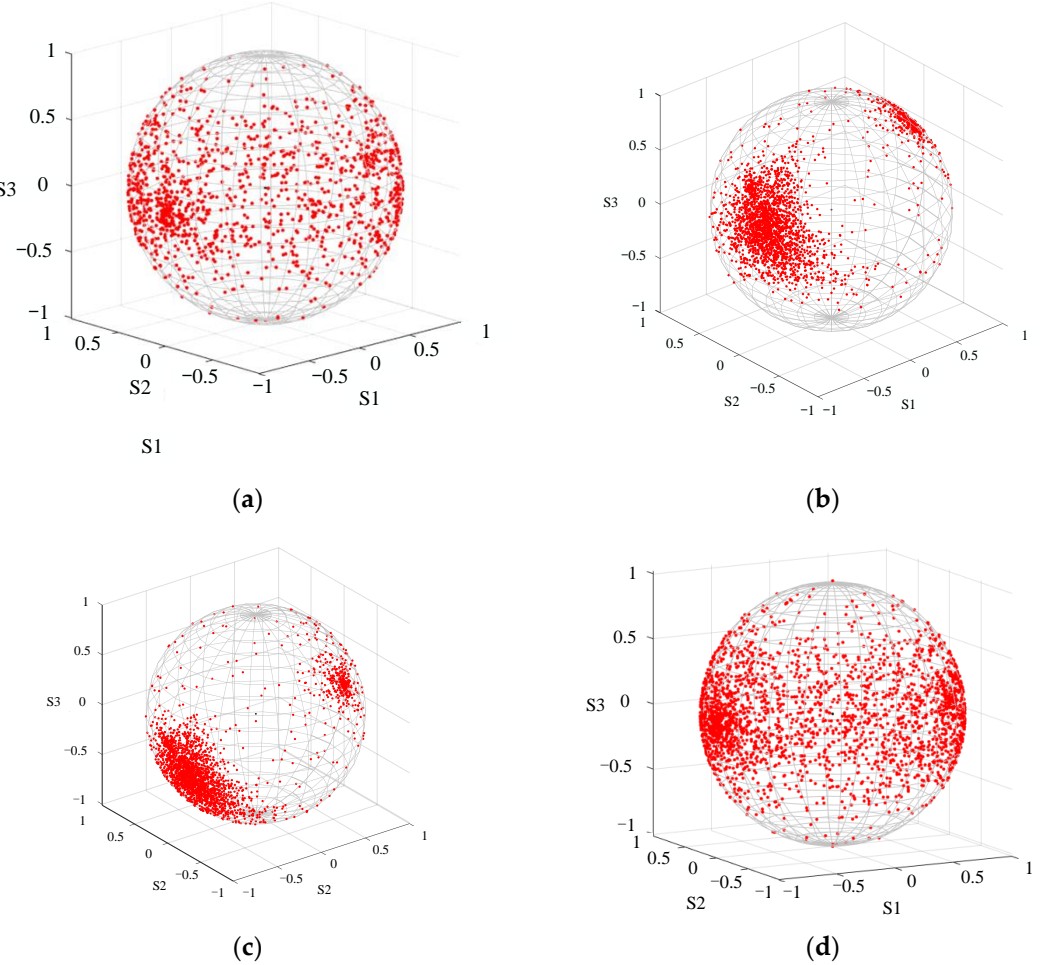

**Figure 12.** Elliptically polarized signal light and three polarization states of local oscillator light: linear polarization, circular polarization, and elliptical polarization. (**a**) Elliptically polarized signal light; (**b**) linearly polarized local oscillator light; (**c**) circularly polarized local oscillator light; and (**d**) elliptically polarized local oscillator light.

The waveform of the intermediate frequency signal collected by the oscilloscope is shown in Figure 13. It can be seen from the figure that when the signal light is in an elliptical polarization state and the local oscillator light is elliptically polarized, the peak-to peak-value of the intermediate frequency signal is 254.8 mV; when the signal light is in an elliptical polarization state and the local oscillator light is circularly polarized, the peak-to-peak value of the intermediate frequency signal is 124.0 mV; and when the signal light is in an elliptical polarization state and the local oscillator light is linearly polarized, the peak-to-peak value of the intermediate frequency signal is 94.4 mV.

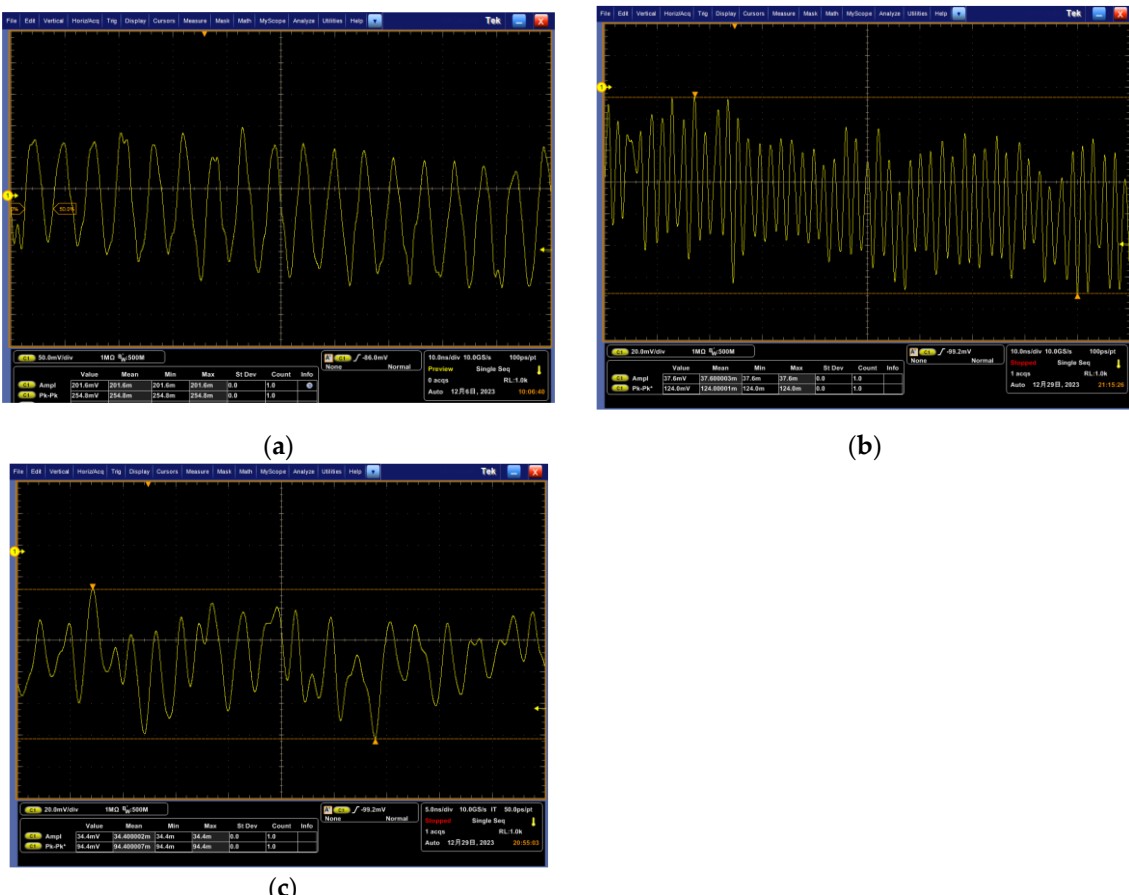

**Figure 13.** Intermediate frequency signal waveforms: (**a**) Intermediate frequency signal diagram when the signal light is elliptically polarized and the local oscillator light is linearly polarized; (**b**) intermediate frequency signal diagram when the signal light is elliptically polarized and the local oscillator light is circularly polarized; and (**c**) intermediate frequency signal diagram when the signal light is elliptically polarized and the local oscillator light is linearly polarized.

The mixing efficiency is higher when the signal light's polarization state is the same as or similar to that of the local oscillator light, as demonstrated in Figures 9, 11 and 13. This is due to the fact that energy conversion and frequency conversion in nonlinear media require the signal light and the local oscillator light to have the same vibration direction and phase relationship during the mixing process. Their vibration directions are more consistent and energy conversion is more efficient when their polarization states are the same or comparable. Conversely, energy conversion efficiency will drop if there is a significant difference between the polarization states of the two.

When the polarization state of the signal light is the same as that of the local oscillator light, we adjust the local oscillator light power to about 2 mW, adjust the signal light power so that the amplitude of the intermediate frequency signal is 450 mv, and measure the power of the signal light under three different states. From Figure 14, it can be seen that

when the signal light is a linearly polarized EGSM beam, the measured output power of the signal light fluctuates around −47 dBm; when the signal light is a circularly polarized EGSM beam, the measured output power of the signal light fluctuates around −52 dBm; and when the signal light is an elliptically polarized EGSM beam, the measured output power of the signal light fluctuates around −43 dBm.

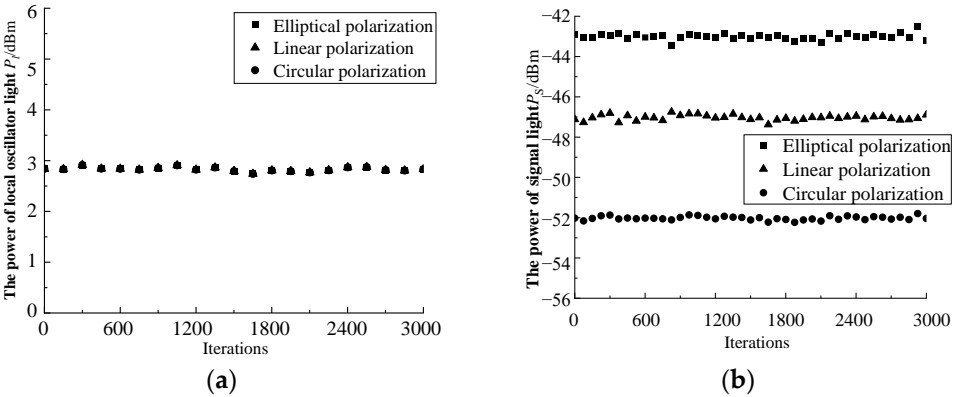

**Figure 14.** Measurement curves of signal light and local oscillator power under different polarization states of EGSM beams. (**a**) Measurement curves of local oscillator power under different polarization states of EGSM beams; (**b**) measurement curves of signal light power under different polarization states of EGSM beams.

The output power of the local oscillator was adjusted to around 2.4 mW and the amplitude of the intermediate frequency signal was measured to be around 450 mV. The signal light power was measured in three different states, as shown in Figure 15. When the signal light is a linearly polarized EGSM beam, the output power of the measured signal light fluctuates around −50 dBm; when the signal light is a circularly polarized EGSM beam, the measured output power of the signal light fluctuates around −54 dBm; and when the signal light is an elliptically polarized EGSM beam, the measured output power of the signal light fluctuates around −45 dBm.

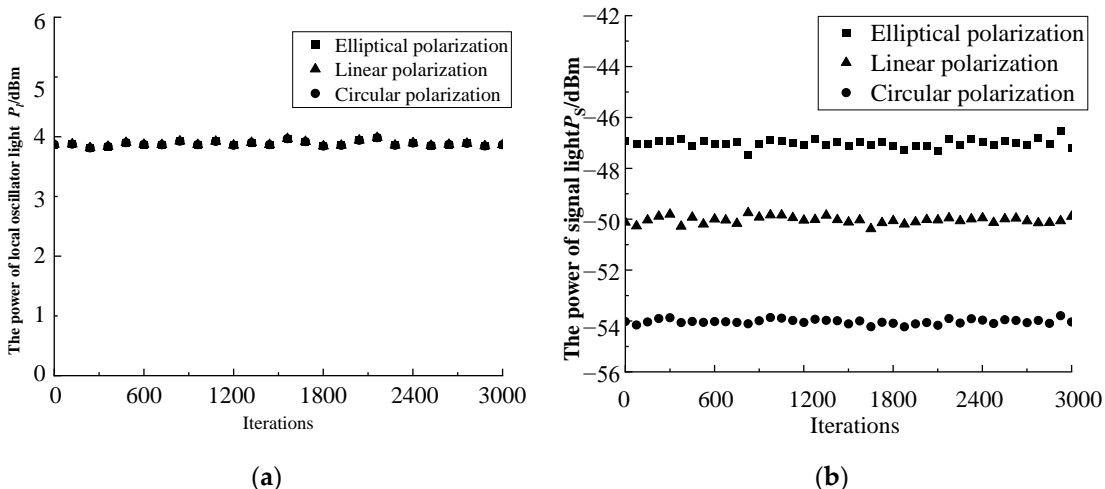

**Figure 15.** Measurement curves of signal light and local oscillator power under different polarization states of EGSM beams. (**a**) Measurement curves of local oscillator power under different polarization states of EGSM beams; (**b**) measurement curves of signal light power under different polarization states of EGSM beams.

From Figure 16, it can be seen that when the local oscillator power is 3 mw, and the signal power is adjusted so that the amplitude of the intermediate frequency signal

is 450 mv, the signal power in three states can be measured. When the signal light is a linearly polarized EGSM beam, the output power of the measured signal light fluctuates around $-54$ dBm; when the signal light is a circularly polarized EGSM beam, the measured output power of the signal light fluctuates around $-55$ dBm; and when the signal light is an elliptically polarized EGSM beam, the measured output power of the signal light fluctuates around $-48$ dBm.

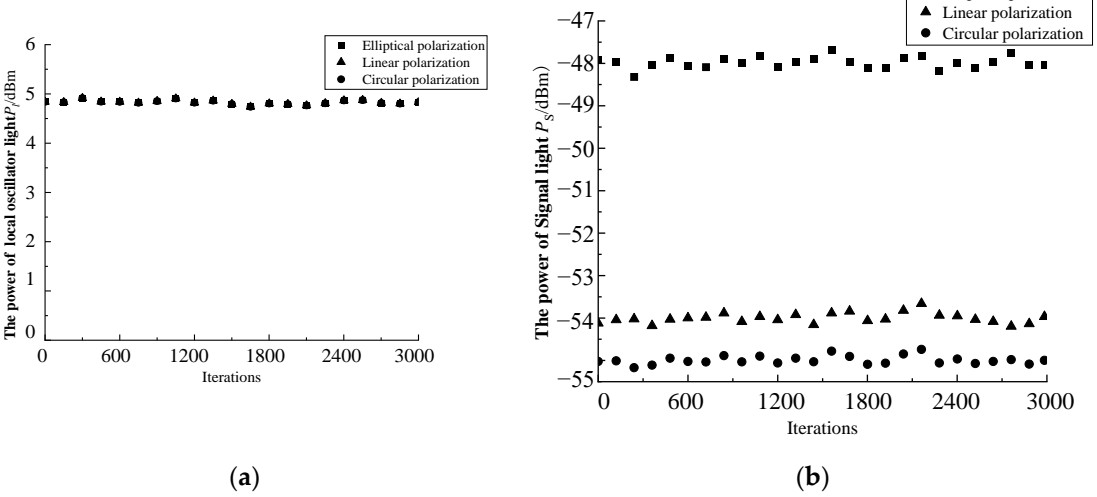

(**a**)                                    (**b**)

**Figure 16.** Measurement curves of signal light and local oscillator power under different polarization states of EGSM beams. (**a**) Measurement curves of local oscillator power under different polarization states of EGSM beams; (**b**) measurement curves of signal light power under different polarization states of EGSM beams.

It is evident from the three scenarios that the detection sensitivity increases with the local oscillator power. Circularly polarized EGSM beams have higher sensitivity when EGSM beams with different polarization states are used for coherent detection, although there may be slight variations in the signal light's power and its deviations when adjusting the polarizer. In the case of EGSM beams, their output power can be affected by the polarization states of the beams. When the polarization states are different, it implies that the electric field vectors of the beams are oriented at different angles, or that they have different circular polarization directions. These differences in polarization states can result in an unequal coupling efficiency and unbalance interactions between the optical components, such as the mixers mentioned earlier. As a result, there may be variations in the power levels of the EGSM beams with different polarization states.

## 4. Conclusions

This study theoretically derived and experimentally verified the relationship between the polarization state and mixing efficiency of EGSM signal light and local oscillator light beams. The results indicate that:

When both the local oscillator light and the signal light are linearly polarized, the degree of coincidence of the polarization directions directly affects their mixing efficiency. From the simulation results, it can be seen that when the angle between the local oscillator vector and the horizontal direction is 45°, and the angle between the signal vector and the horizontal direction is 45°, the mixing efficiency is the highest, and the efficiency varies with the cosine function of the polarization direction angle between the two. Circularly polarized EGSM beams can be seen as the combination of two linearly polarized light signals with a phase difference of 90° in the vertical direction, with two components of equal amplitude but perpendicular phases. If the signal light is circularly polarized, a higher mixing efficiency can be maintained. Elliptically polarized EGSM beams can be regarded as a general case of linear polarization and circular polarization, consisting of

two orthogonal components with unequal amplitudes and a phase difference of $90°$. The mixing efficiency depends on the degree of matching between these two components and the corresponding component of the local oscillator light. When the main axis of the ellipse is aligned with the polarization state of the local oscillator, the mixing efficiency is higher, but the greater the deviation of the ellipse shape from the polarization state of the local oscillator, the lower the efficiency. When both the signal light and the local oscillator light are in the same polarization state (whether that is linear polarization, circular polarization, or elliptical polarization), the mixing effect is better.

**Author Contributions:** Conceptualization, Y.M. and X.K.; methodology, J.L., Y.M., X.K. and M.H.; software, Y.M.; validation, J.L., Y.M. and M.H.; formal analysis, X.K.; investigation, Y.M. and X.K.; resources, X.K.; data curation, M.H.; writing—original draft preparation, Y.M.; writing—review and editing, Y.M.; visualization, Y.M.; supervision, X.K.; project administration, X.K.; funding acquisition, X.K. All authors have read and agreed to the published version of the manuscript.

**Funding:** Funding was received from the following: the Key Industrial Innovation Chain Project of Shaanxi Province [grant number 2017ZDCXL-GY-06-01]; the General Project of National Natural Science Foundation of China [grant number 61377080]; the Xi'an Science and Technology Plan (22GXFW0115); the Scientific Research Team of Xi'an University (D202309); and the Xi'an Science and Technology Program Fund (2020KJRC0083).

**Institutional Review Board Statement:** This study did not require ethical approval.

**Informed Consent Statement:** This study did not require ethical approval.

**Data Availability Statement:** The data that support the findings of this study are available from the corresponding author upon reasonable request.

**Conflicts of Interest:** The authors declare no conflicts of interest.

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
