# Peer review of "Experimental Study on Partially Coherent Optical Coherent Detection"

_photonics, doi:10.3390/photonics11020160_

Round 1

Reviewer 1 Report

Comments and Suggestions for Authors

This article analyzes the relationship between local oscillator light EGSM beams of different polarization states, signal light EGSM beams and mixing efficiency from both theoretical and experimental aspects. It is concluded that the matching status of polarization states of signal light and local oscillator light affects the mixing efficiency, which benefits the design of free space laser communication systems. But there are currently some questions that need to be answered by the author.

1There is an issue with non-standard units in this article. For example, "mv" should be changed to "mV"; "mw" should be changed to "mW".

2“Bij” marked 79 in this article does not comply with the specification.

3Please indicate in detail the references cited for the formulas in the text, such as formula (1).

4The font size of formula (9) is not consistent with that of other formulas.

5The angle q appears in this article, but the specific expression of q is not found. Please elaborate on q.

6The article does not elaborate on the content of Figure 10. The author is requested to provide additional explanations. Similarly, Figure 12 has the same problem.

7Please specify the power of the signal laser at numbers 319 and 333 in the text to facilitate comparison of the results.

8The format of the paragraph marked 207 should be consistent with the above.

9There is a misrepresentation in the titles of Figure 14(a) and Figure 14(b). For example, Figure 14(a) shows the power of local oscillator light under different iteration numbers, but the title of Figure 14(a) is the opposite. Similarly, there are errors in the subfigure titles of Figures 15 and 16.

10There are problems with the citation format of some references in this article, such as the page number of Document 12, Document 14 should be published on pages 1370-1380 of Volume 51, Issue 5, and the journal name of Document 16 is incorrectly stated. Authors are asked to check the citation format of the document to meet the standards for journal publication.

Comments on the Quality of English Language

Minor English editing is required.

Reviewer 2 Report

Comments and Suggestions for Authors

In the presented manuscript, a detailed technique for calculating the superposition (summation) of two electromagnetic waves is presented, basically considering the influence of their polarizations on the resulting wave. Mathematically, things are presented acceptably. The work, in its generality, can be practically useful in this type of analysis. Questions can be raised - as a scientific publication, I think the scientific novelty of the work should be more explicitly discussed and presented. In general, in this presentation, one is left with the impression of presenting a mathematical technique of implementing known basic things. In the work, the new scientific points in the development - specification of the basic scientific novelties - should be highlighted and emphasized, as far as the consideration leaves the impression of a mathematical technique of application of known approaches. Also, one should cite other works that highlight the same issue of calculating the interference of EGSM waves with different polarizations. I think it is also appropriate to discuss the results of the following papers, for example:

1. Hema Roychowdhury, Olga Korotkova Realizability conditions for electromagnetic Gaussian Schell-model sources, Optics Communications, Volume 249, Issues 4–6, 15 May 2005, Pages 379-385,

3. Biling ZhangYonggen XuYouquan DanXueru DengZhengquan Zhao , Beam spreading and M2-factor of electromagnetic Gaussian Schell-model beam propagating in inhomogeneous atmospheric turbulence, Original research article, Optik, Volume 149, November 2017, Pages 398-408

4. Xianyang Yang, Wenyu Fu, Xuehua Hu, Xue Li, Properties of an electromagnetic twisted Gaussian Schell-model array beam propagating in anisotropic atmosphere turbulence, Optica Applicata, Vol. LII, No. 4, 2022, DOI: 10.37190/oa220413

5. Min Yao, Yangjian Cai, Halil T. Eyyuboğlu, Yahya Baykal, and Olga Korotkova, Evolution of the degree of polarization of an electromagnetic Gaussian Schell-model beam in a Gaussian cavity, Optics Letters, Vol. 33, Issue 19, pp. 2266-2268  (2008), https://doi.org/10.1364/OL.33.002266

6. Yuanhang Zhao , Yixin Zhang, Qiu Wang, Average Polarization of Electromagnetic Gaussian Schell-Model Beams through Anisotropic Non-Kolmogorov Turbulence, Optical Communications, Radioengineering, Vol. 25, No. 4, December 2016, 652 – 657.

In clarifying the question posed to highlight the underlying scientific novelty, the work will have the form of a proper scientific publication. Although the technique used is presented, the selection and highlighting of the scientific novelty needs to be definitely better brought out. Otherwise, the work leaves the impression of a technical development in mathematical aspect.

Some of the noticed problems are noted in the attached file.

Comments on the Quality of English Language

In some parts of the manuscript the English is difficult to understand. Difficulties are also related to the incorrect formulation of sentences, such as separating one from another with semicolons. Review the formation and presentation of shorter sentences for easier understanding.

Reviewer 3 Report

Comments and Suggestions for Authors

In this manuscript, Liang et al. explore the impact of partially coherent light polarization on mixing efficiency, and examine the impact of partially coherent EGSM beams on system performance with different polarization states. They derived that when the polarization state of the signal light is similar or consistent with the polarization state of the local oscillator light, a better mixing effect is achieved. The results were verified numerically and experimentally. The results are good, but the article writing is so poor. The formula format in the article is inconsistent; they do not cite the order of the relevant graphs, when they describe the results of the graphs, such as Figs. 14-16. English is not good and there are many typos. Please carefully check the whole manuscript and give the reasonable responses to the following comments. After that, I would like to recommend it to be accepted by your journal.

1.      There are many typos, such as “Shaanxi” should be “Shanxi” in the affiliation, the comma should be replaced by the period in line 80, “Where” should be replaced by “where” in line 81 et al.

2.      The sections of abstract and conclusion should be concise.

3.      In the introduction, the authors proposed that optical mixers in coherent optical communication systems are sensitive to the polarization states of local oscillator light and signal light. The optical mixing efficiency will be affected by changes in the polarization state. If the two polarization states are different, the performance of the entire communication system will be adversely affected. The authors should give more details about how their impact was felt.

4.      In Figure 1, three thumbnails are a bit blurry.

5.      Please give more details about the coherent detection principle, for the general readers.

6.      Please explain the different output power of the EGSM beams with the different polarization states.

Comments on the Quality of English Language

It should be polished.

Round 2

Reviewer 2 Report

Comments and Suggestions for Authors

The authors have made an effort to take the comments into account. Once again to be reviewed for possible inaccuracies of expression in sentences and syntax errors .

Reviewer 3 Report

Comments and Suggestions for Authors

none